# Enhanced Enzymatic Hydrolysis of *Pennisetum alopecuroides* by Dilute Acid, Alkaline and Ferric Chloride Pretreatments

**DOI:** 10.3390/molecules24091715

**Published:** 2019-05-02

**Authors:** Shangyuan Tang, Chunming Xu, Linh Tran Khanh Vu, Sicheng Liu, Peng Ye, Lingci Li, Yuxuan Wu, Mengyu Chen, Yao Xiao, Yue Wu, Yining Wang, Qiong Yan, Xiyu Cheng

**Affiliations:** 1College of Life Sciences and Bioengineering, School of Science, Beijing Jiaotong University, Beijing 100044, China; 16121638@bjtu.edu.cn (S.T.); 16292010@bjtu.edu.cn (S.L.); 17121611@bjtu.edu.cn (P.Y.); 18121614@bjtu.edu.cn (L.L.); 15272020@bjtu.edu.cn (Y.W.); 18272032@bjtu.edu.cn (M.C.); 17271250@bjtu.edu.cn (Y.X.); 18272052@bjtu.edu.cn (Y.W.); 17272018@bjtu.edu.cn (Y.W.); 2Key Laboratory of Cleaner Production and Integrated Resource Utilization of China National Light Industry, Beijing Technology and Business University, Beijing 100048, China; xucm@th.btbu.edu.cn; 3Faculty of Chemical and Food Technology, Ho Chi Minh City University of Technology and Education, No. 1 Vo Van Ngan Street, Linh Chieu Ward, Thu Duc District, Ho Chi Minh City 71307, Viet Nam; linhvtk@hcmute.edu.vn

**Keywords:** *Pennisetum alopecuroides*, dilute alkaline pretreatment, ferric chloride pretreatment, enzymatic hydrolysis

## Abstract

In this study, effects of different pretreatment methods on the enzymatic digestibility of *Pennisetum alopecuroides*, a ubiquitous wild grass in China, were investigated to evaluate its potential as a feedstock for biofuel production. The stalk samples were separately pretreated with H_2_SO_4_, NaOH and FeCl_3_ solutions of different concentrations at 120 °C for 30 min, after which enzymatic hydrolysis was conducted to measure the digestibility of pretreated samples. Results demonstrated that different pretreatments were effective at removing hemicellulose, among which ferric chloride pretreatment (FCP) gave the highest soluble sugar recovery (200.2 mg/g raw stalk) from the pretreatment stage. In comparison with FCP and dilute acid pretreatment (DAP), dilute alkaline pretreatment (DALP) induced much higher delignification and stronger morphological changes of the biomass, making it more accessible to hydrolysis enzymes. As a result, DALP using 1.2% NaOH showed the highest total soluble sugar yield through the whole process from pretreatment to enzymatic hydrolysis (508.5 mg/g raw stalk). The present work indicates that DALP and FCP have the potential to enhance the effective bioconversion of lignocellulosic biomass like *P. alopecuroides*, hence making this material a valuable and promising energy plant.

## 1. Introduction

With the growth of vehicles and an over-reliance on fossil fuels in industrial development, biofuel production from non-food crop feedstocks (e.g. agricultural residues, forest residues and industrial wastes) has drawn considerable attention [1,2,3,4,5,6,7,8]. In China, anaerobic digestion of organic wastes has been successfully used for biogas production both domestically, as well as at a larger scale [2,3,4,5,6]. Different biomass substrates, such as cornstalk, rice straw, wheat straw and pine foliage, have also been investigated for the production of bioethanol [9,10,11,12]. In addition to these lignocellulosic biomass wastes, *Pennisetum alopecuroides* is a potential prolific renewable herbaceous plant that is widely distributed in many provinces of China. Some *Pennisetum* species are cultivated as an important feedstock for the animal feed industry in north China, northeast China, southwest China and the middle-lower Yangtze Plain [8]. The annual dry matter yield of *Pennisetum* grass ranges from 40 to 50.2 t/ha, which is much higher than the yields of sugarcane and corn (13~21 t/ha) and comparable to that of miscanthus (27~38 t/ha) [9,10]. While *Miscanthus* species and some other energy plant candidates (e.g., switchgrass) have attracted widespread interest in the bio-energy field, studies on *P. alopecuroides* have been relatively limited. Given its rapid growth, high yield, high adaptability, low cost and environmentally benign production, *P, alopecuroides* was selected as a substrate in the current study to evaluate its application prospects in biofuel production.

The recalcitrant structure of lignocellulosic biomass is the main constraint of its bioconversion [13,14,15,16,17,18]. Varied pretreatment strategies such as chemical methods (e.g., acid, bases, salts and solvents), physico-chemical processes (e.g., steam explosion, liquid hot water (LHW) and ammonia fiber expansion (AFEX) and biological methods have been developed in attempts to remove hemicellulose and/or lignin from lignocellulosic wastes and reduce the crystallinity of cellulose [13,19,20,21,22,23,24]. It is widely accepted that efficient pretreatment should avoid the use of expensive chemicals, improve fiber reactivity and maximize the recovery/formation of fermentable sugars, avoid formation of enzyme inhibitory byproducts, preserve cellulose and hemicellulose fractions that are easily accessible to hydrolysis enzymes and minimize energy requirements [11,12,13,14]. However, no single strategy could efficiently meet all these criteria due to the variations in material characteristics. The chemical pretreatment of lignocellulosic materials has been widely employed in many pilot and large-scale cellulosic ethanol plants because it is ideal for low-lignin materials and has high reactivity at mild conditions [12,13,14]. A chemical method is hence a suitable pretreatment strategy for *P. alopecuroides,* a low-lignin material.

Among several chemical methods, dilute acid pretreatment is most commonly used, due to its advantages in cost and process severity [13,14]. One major limitation of acid pretreatment is its requirement of corrosion-resistant reactors [13]. On the other hand, corrosion problems and sugar degradation are less severe in alkaline processes than in acid pretreatment. Alkaline pretreatment is also effective in delignifying the biomass [7,20,22,24]. A mild alkali concentration (<4% *w/w*) favors enzymatic hydrolysis especially for low-lignin materials [13,14]. Previous studies also showed that pretreatment of lignocellulosic biomass using Lewis acids such as FeCl_3_ enhances digestibility of biomass, producing reusable solubilized hemicellulose [25,26,27]. Operating at milder temperature conditions (120 °C vs 160~260 °C in steam explosion and LHW pretreatments) in the current study means that dilute acid pretreatment (DAP), dilute alkaline pretreatment (DALP) and ferric chloride pretreatment (FCP) reduce energy consumption and the formation of enzyme inhibitory byproducts [11]. Up to date, DAP, DALP and FCP have been used on a wide range of low-lignin biomasses ranging from wood (cedar, pine, hemlock) to agricultural residues (corn stover, wheat and barley straw, switchgrass, *Miscanthus*) [18,19,20,21,22,23,24]. To our best knowledge, the effects of these methods on the biodigestibility of *P. alopecuroides* has not been systematically studied to identify the ideal pretreatment process or to evaluate the potential of *P. alopecuroides* biomass in the bioconversion industry. Moreover, some previous studies have also shown that effective removal of lignin and/or hemicellulose in acid/alkaline pretreatments did not result in a significant increase in reducing sugar yields (only 91.4−92 mg/g) [19,21]. These results indicate that the exact roles of different pretreatments in the improvement of hydrolysis efficiency were complicated, thereby necessitating further research to better understand the mechanism.

In the present study, three chemical pretreatments including DAP, DALP and FCP were systematically investigated to develop an efficient pretreatment strategy for enhancing enzymatic hydrolysis of *P. alopecuroides* biomass. The composition and microstructure of substrates in response to these pretreatments were investigated to better understand the exact roles of each pretreatment in changing biomass recalcitrance and subsequent enzymatic hydrolysis enhancement. The fermentable sugar production from the pretreated *P. alopecuroides* biomass was also studied to evaluate its application prospects for biofuel production.

## 2. Results and Discussion

### 2.1. Effect of Different Pretreatments on Biomass Composition

The pretreatment process decreases the recalcitrance of lignocellulosic substrates by removing lignin and hemicellulose components, thereby exposing cellulose to the hydrolysis enzyme [28,29]. The *P. alopecuroides* samples were subjected to different pretreatments, including DAP, DALP and FCP. The solid yield and compositional change of the stalk samples are important indices to evaluate the effectiveness of their pretreatments. As shown in Table 1, the dry matter retained after different pretreatments was about 53.3−58.2%. A *P. alopecuroides* sample pretreated by DALP had a higher solid yield as compared with those pretreated with DAP and FCP. The weight loss of *P. alopecuroides* biomass could be attributed to the solubilization of its components into the aqueous solution. The higher solid yield (or lower weight loss) indicate that less lignocellulosic components were converted into soluble substances. In comparison to DALP, DAP and FCP thereby gave higher soluble sugar concentrations (86.7 mg/g raw stalk (RS) vs 112.2 mg/g RS and 193.4 mg/g RS, respectively).

Compositional analysis showed that the weight loss of the stalk samples was mainly due to hemicellulose hydrolysis by both DAP and FCP (Table 1). A previous study also reported that soluble sugars predominantly originated from the hemicellulose fraction of lignocellulosic substrates [23]. The dilute acid pretreatment and FeCl_3_ pretreatment can markedly solubilize hemicellulose into monomeric sugars and soluble oligomers [23,27]. Results in Table 1 indicate that DALP significantly removed both lignin and hemicellulose, while DAP and FCP preferentially removed hemicellulose. Samples pretreated by DALP had the lowest lignin content of 11.7%. A similar delignification effect was also observed in alkaline pretreatment and microwave-assisted alkaline pretreatment processes for different stalk wastes [6,15,30,31].

### 2.2. Effect of Different Pretreatments on Cell Structure

Cell structure changes of the stalk samples pretreated by DAP, DALP and FCP were also studied by a scanning electron microscopy (SEM) at the same magnification (× 500) (Figure 1).

As shown in Figure 1a, the untreated *P. alopecuroides* sample had a smooth and intact surface with an unchanged fibrous structure organization. On the other hand, the cell walls of the DAP sample were obviously destroyed. It can be seen in Figure 1b that the rectangular cell wall boundaries became blurry to some extent, indicating that a portion of cellulosic components could have been removed by DAP. The cell surfaces of the FCP sample were also not smooth, and partly broken (Figure 1d). This clearly verifies that the removal of lignocellulosic components such as hemicellulose by FCP resulted in morphological surface modification. Stronger morphological changes were observed from the DALP sample. As can be seen in Figure 1c, the surfaces and cell walls of the DALP sample were significantly destroyed and distorted. Enhanced destruction of plant cell wall was also found in previous studies with microwave-assisted acid/alkaline pretreatments of lignocellulosic wastes [22,23,32]. The present study suggests that the strong delignification effect and partial removal of hemicellulose by DALP remarkably enhances the destruction effect to cell wall structure (Table 1 and Figure 1). The formation of cracks, fragments and the distortion of cell structure induced by DALP could lead to the formation of many reactive sites on the biomass surface. Such changes increasingly benefit the accessibility of stalk samples to enzymes and bacteria, and consequently accelerate the subsequent bioconversion process.

### 2.3. Effect of Different Pretreatments on Enzymatic Hydrolysis

Enzymatic hydrolysis was carried out to evaluate the enhancement of biodegradability of *P. alopecuroides* biomass pretreated with DAP, DALP and FCP. As shown in Figure 2, the untreated sample displayed its lowest reducing sugar yield after 72 h of enzymatic hydrolysis (134.8 mg/g raw stalk). Meanwhile, samples pretreated with 3.0% H_2_SO_4_ solution and 3.2% FeCl_3_ solution under 121 °C for 30 min exhibited varied increases in enzymatic hydrolysis efficiency. The reduced sugar yields of the pretreated samples by DAP and FCP reached 258.3 mg/g pretreated stalk (PS) and 293.7 mg/g PS, increasing by 92% and 118% as compared with the untreated samples, respectively. DALP appeared to be the most efficient pretreatment technique by which to obtain the highest reducing sugar yield among all tested pretreatment methods (669.7 mg/g PS) (Figure 2).

The highly ordered crystalline structure of cellulose fibrils, the presence of lignin polymer and the recalcitrant cellulose–hemicellulose–lignin network structure in lignocellulosic biomass severely hindered enzymatic hydrolysis [33,34]. As shown in Table 1 and Figure 1, decreases in hemicellulose content and changes in cell structure, emerging after acid pretreatments (especially FCP), were responsible for the significant increase in reducing sugar yield. It is obvious that the pretreated sample by DALP, in which much more lignin components were removed, showed a much higher reducing sugar yield than those of DAP and FCP samples. The connection between these results strengthens a hypothesis that lignin removal is relatively important for improving enzymatic hydrolysis. This is because delignification not only reduces the adsorption of cellulase onto lignin, but also produces higher cellulose substrate content, consequently making cellulose more accessible to enzymes [30,31,33,34]. A stronger destruction of cell wall and higher increase of cellulose accessibility in DALP samples also produced a positive impact on enzymatic hydrolysis, affirmed by SEM observation (Figure 1).

### 2.4. Effect of H_2_SO_4_/NaOH/FeCl_3_ Concentrations on Biomass Composition

The effects of H_2_SO_4_/NaOH/FeCl_3_ concentrations in different pretreatment processes on the chemical characteristics of pretreated samples were further studied. The corresponding acid, alkali and FeCl_3_ concentrations in DAP, DALP and FCP were selected based on previous studies [7,22,23,27]. In DAP, as the H_2_SO_4_ concentrations increased from 1.0% to 4.0%, the solid yields decreased from 59.4% to 52.6%. The reduction in solids was mainly ascribed to the degradation of lignocellulosic components. As can be seen in Table 2, the corresponding hemicellulose contents decreased from 16.9% to 12.2%, consistent with the decrease of solid yields. Removed cellulosic components were partially converted into soluble sugars, and the corresponding sugar concentrations in the aqueous phase significantly increased from 40.0 mg/g RS to 119.3 mg/g RS (Table 2). Due to the removal of hemicellulose, cellulose contents increased.

In the case of FCP, severe pretreatment conditions (3.2% and 4.8% FeCl_3_) also effectively removed cellulosic components, resulting in lower solid yields of 53.1~55.9% as compared with those of less severity conditions (0.8~1.6% FeCl_3_) (Table 2). The significant decrease in hemicellulose content and the slight increase in cellulose content suggest that hemicellulose was more easily degraded than cellulose during FCP pretreatment [7,18]. The corresponding soluble sugar content in the aqueous phase also increased from 90.2 mg/g RS to 200.2 mg/g RS (Table 2). Due to the acidic condition provided by H_2_SO_4_ and FeCl_3_ during DAP and FCP, the generated soluble sugars could be dehydrated to furfural and 5-hydroxymethylfurfural (5-HMF) [23,27]. Although the solid yields obtained by DAP and FCP were similar, the soluble sugar concentrations in FCP samples were much higher than those observed in DAP samples (193.4~200.2 mg/g RS vs 112.2~119.3 mg/g RS), indicating that more generated soluble sugars may have been further converted into other byproducts by H_2_SO_4_ [23,27].

The mass loss observed after acid pretreatments can be mainly attributed to the removal of hemicellulose components [23,27]. Acid pretreatments such as DAP and FCP randomly break glycosidic bonds, removing hemicellulose while improving the cellulose content of lignocellulosic biomass and consequently increasing the accessibility of cellulose to hydrolytic enzymes [28,29]. On the other hand, alkaline pretreatments produce nucleophilic attacks that break the lignin structure, solubilizing lignin fragments or hemicellulose from α-O-4 linkages [35,36,37]. Results in Table 2 show that relatively higher solid yields of 53.9~74.6% were observed in DALP samples. As the NaOH concentrations increased from 0.4% to 1.2%, the hemicellulose contents decreased from 22.0% to 14.8%, and cellulose contents significantly increased from 53.3% to 68.0%. The corresponding soluble sugar concentrations also increased from 41.8 mg/g RS to 107.3 mg/g RS (Table 2). It should be noted that DALP, with 1.0~1.2% NaOH, more significantly reduced the lignin contents of biomass to 10.2~11.7% as compared with those achieved by DAP and FCP (Table 2).

### 2.5. Effect of H_2_SO_4_/NaOH/FeCl_3_ Concentrations on Enzymatic Hydrolysis

The effects of different H_2_SO_4_/NaOH/FeCl_3_ concentrations on the enzymatic hydrolysis of *P. alopecuroides* biomass were also investigated. As shown in Figure 3, samples pretreated with DAP, DALP and FCP had much higher reducing sugar yields after enzyme hydrolysis, 1.4~5.5 times higher than that of the untreated sample (134.8 mg/g). In particular, the sample pretreated with 1.0% H_2_SO_4_ solution under 121 °C for 30 min exhibited an obvious increase in enzymatic hydrolysis efficiency, and the corresponding reducing sugar yield reached 258.1 mg/g PS. Increased acid concentrations showed a beneficial effect on cellulosic component removal and subsequent enzymatic hydrolysis. The reducing sugar yield reached a maximum value of 336.4 mg/g PS when *P. alopecuroides* biomass was pretreated with 4.0% H_2_SO_4_. Similar improvements were observed during the FCP process. As can be seen in Figure 3, the reducing sugar yield of the sample pretreated with 0.8% FeCl_3_ solution was 192.0 mg/g PS. The highest reducing sugar yield of 279.3 mg/g PS was obtained when FeCl_3_ concentration was increased to 4.8%.

As can be seen in Figure 3, the enzymatic hydrolysis efficiency of DALP samples was the highest among the untreated sample and all pretreated samples. At the lowest NaOH concentration of 0.4%, the reducing sugar yield of the DALP sample reached 296.1 mg/g PS, comparable to that obtained in samples pretreated with 3% H_2_SO_4_ and 4.8% FeCl_3_. DALP with higher NaOH concentrations (0.6~1.2%) resulted in a significant improvement in hydrolysis performance. Compared to all tested samples, the highest reducing sugar yield was 744.4 mg/g PS using 1.2% NaOH. It should be also noted that some undesirable enzyme-inhibiting byproducts (mainly phenolic compounds, furans and organic acids) might be formed under certain pretreatment processes, especially those conducted at severe conditions (e.g., steam explosion, LHW, chemical treatments, 160~240 °C, 5~45 min, pretreatment severity factor (log *R_0_*) of 2.8~4.8) [11,38,39,40]. However, tested pretreatment strategies were still effective because cellulase was not inhibited in the subsequent enzymatic hydrolysis of the pretreated biomass, as illustrated by high reducing sugar yields of the pretreated samples (Figure 3). These results also confirmed the potential of these chemical pretreatment methods. The low inhibiting effect could be mainly attributed to the milder operation conditions (low acid/alkaline/FeCl_3_ concentrations, 120 °C, 30 min, pretreatment severity factor (log *R_0_*) of 2.1), which can lead to lower levels of byproducts [11,20,41,42,43]. In addition, most byproducts, such as the phenolics formed by the NaOH pretreatment, were retained in the pretreated liquors, resulting in a substantially lower level of phenolics in the enzyme-saccharified hydrolysates [44].

The effects of different pretreatment strategies on removing the physical barrier of lignin, reducing the dense crystalline structure and coating effect of hemicellulose and enhancing the accessibility of the pretreated biomass to hydrolytic enzymes were relatively complicated [11,12,45,46]. Results clearly indicate that both DAP and FCP were effective at removing hemicellulose components and destroying cell wall structure. The hemicellulose removal in DAP and FCP exhibited a positive relationship with the reducing sugar yields during the subsequent enzymatic hydrolysis (Table 2 and Figure 3). These results further support the observation that under acid/Lewis acid pretreatments, cellulose saccharification is linearly proportional to the amount of hemicellulose (mainly xylan) removed, since hemicellulose removal helps increase cellulose accessibility [27,45]. The reducing sugar yields (279.3~336.4 mg/g PS) and conversion ratios (34.6~44.5%) by DAP and FCP were comparable to those observed in the pretreated bamboo (77 mg/g PS), rice grass (457 mg/g PS) and pine foliage (588 mg/g PS) pretreated with 1~2% H_2_SO_4_ at 121 °C for 60 min [19,20,21]. Notably, the simultaneous removals of lignin and hemicellulose were also positively related with the reducing sugar yields of the DALP samples (Table 2 and Figure 3). Despite lower hemicellulose removal, DALP provided a much higher delignification (Table 2) and reducing sugar recovery as compared with DAP and FCP (744.4 mg/g PS vs 279.3~336.4 mg/g PS, respectively). In addition to the obvious impact on delignification, significant enhancements on fermentable sugar releases were also observed in different biomass samples pretreated with NaOH under similar conditions, including bamboo, pine foliage, *Pennisetum purpureum*, wheat straw and *Eucalyptus* (324~629 mg/g PS) [17,20,34,44]. These results indicate that effective lignin removal, as well as cellulose swelling induced by DALP, appeared to be more important factors in decreasing biomass recalcitrance and increasing enzymatic digestibility as compared with hemicellulose removal in DAP and FCP.

### 2.6. Analysis of Mass Balance and Prospects for P. alopecuroides

A promising pretreatment method not only enables the ability to obtain readily digestible substrates, but also maximizes the total yield of fermentable sugars. Table 3 shows the mass balance of stalk samples under different pretreatments and subsequent enzymatic hydrolysis. The highest reducing sugar yields in the aqueous phases under the optimal DAP, DALP and FCP conditions were 119.3 mg/g RS (DAP-4% H_2_SO_4_), 107.3 mg/g RS (DALP-1.2% NaOH) and 200.2 mg/g RS (FCP-4.8% FeCl_3_). The solid yields under the above conditions were 52.6%, 53.9% and 53.1%, respectively. The corresponding reducing sugar yields of enzymatic hydrolysis of the DAP, DALP and FCP samples were 336.4, 744.4 and 279.3 mg/g PS, respectively. Based the above data, the total soluble sugar yields from both pretreatment and enzymatic hydrolysis processes were 296.2, 508.5 and 348.6 mg/g RS, respectively. DAP had the lowest sugar recovery, partly because of its low enzymatic hydrolysis efficiency. Another reason was that cellulosic components, once removed, may be further converted into other byproducts, and thus the sugar yield from the aqueous phase during DAP was not high despite of its high weight loss. Among experimental conditions tested, DALP yielded the highest total sugar recovery through the whole process from pretreatment to enzymatic hydrolysis, while FCP indicated the most efficient method to recover soluble sugars only at pretreatment stage. 

The fermentable sugar recovery from the *P. alopecuroides* sample pretreated by DALP was comparable to those obtained in the enzymatic hydrolysis of different lignocellulosic biomasses, as well as *Pennisetum* grass species [9,17,19,20,34,46,47]. As shown in Table 4, the maximum released sugar yields of 362.3~629 mg/g PS were obtained by acid or alkaline pretreatments of various biomass wastes such as wild rice grass, pine foliage, *Eucalyptus* and bamboo [19,20,34,46]. The optimal reducing sugar yield was only 146.9 mg/g pretreated elephant grass (*P. purpureum*) pretreated with 1.5% NaOH at 121 °C for 60 min [17]. Total fermentable sugar yields of 324~537 mg/g Napier grass (*P. purpureum*) were reported for samples pretreated with 2% Ca(OH)_2_ or 2% NaOH at 121 °C for 60 min [47]. In this study, the yield obtained so far on *P. alopecuroides* (pretreated with 1.2% NaOH at 121 °C for 30 min) was 744.4 mg/g PS (508.5 mg/g RS, conversion ratio of 85.4%), opening a potential avenue for efficient biofuel production from *Pennisetum* grass.

Results obtained in this study clearly illustrated that digestibility of *P. alopecuroides* biomass were significantly improved by DALP, hence this pretreated biomass could be used in bioethanol or biogas production [1,12,13,14]. In fact, the DALP sample could be directly fermented by mixed microorganisms for biogas production, hence omitting the enzymatic hydrolysis step using expensive cellulase [1,3,4,5]. The techno-economic feasibility of the DALP sample integrated with biogas production, in which energy recovery ranges from 50% to 85%, has already been proven in several industrial biogas plants in China [1,48,49].

The current pretreatment strategies can also be integrated with ethanol fermentation. An energy recovery of 37% was observed in full-scale ethanol production using starch substrates [49]. Lignocellulosic bioethanol production is also energetically sustainable based on heat/electricity production from fermentation residues, while greenhouse gas (GHG) emissions are decreased by 50~93% in this process [50]. It should be noted that the industrial application of lignocellulosic bioethanol production was still limited to some extent due to the high cost of cellulase (about USD 0.50 per gallon ethanol, accounting for 20%–30% of total costs [51]). However, recent techno-economic analysis indicated that the evaluated minimum ethanol selling price (MESP) decreased from USD 4.58 per gallon to USD 1.91~2.46 per gallon, which comes close to the market price of ethanol (USD 2.50~3.10 per gallon) [52,53,54,55]. Additionally, many studies have also been conducted to further resolve the bottleneck of enzymatic hydrolysis in lignocellulosic bioethanol production as follows: 1) improving production/activity of cellulase using mutagenesis, co-culturing and heterologous gene expression of cellulases; 2) reusing enzymes by immobilization; and 3) process optimization and integration (e.g., simultaneous saccharification and fermentation processes, cost-effective pretreatment, etc.) for reducing the cost [12,13,14]. These research avenues are undoubtedly making lignocellulosic bioethanol more economically viable.

It should be noted that due to limited farmland resources in China, planting energy crops on available marginal land, which is estimated to be about 5.5 million ha, is regarded as one of the most promising choices for the production of biofuel feedstocks [56]. If 20% of the marginal land area is used for planting *P. alopecuroides* and the dry biomass yield is about 30 t/ha, theoretically about eight million tons of cellulosic ethanol can be produced annually, assuming a 12% biomass moisture content and an ethanol yield of 264 kg/t of dry biomass [52]. This estimated potential yield would almost reach the 2020 ethanol target (10 million tons per year) in China.

## 3. Materials and Methods

### 3.1. Materials

Wild *P. alopecuroides* was manually collected from Fujian, China, and dried in the sun. After that, *P. alopecuroides* samples were dried in an oven at 60 °C for at least 24 h to a constant weight and then milled to pass through a 20-mesh sieve using a plant miller. The main composition of stalks (on a dry weight basis) was as follows: cellulose 41.8%, hemicellulose 28.7% and lignin 17.5%.

### 3.2. Pretreatment Process

Dried samples were added to glass bottles containing 1.0~4.0% (w/v) sulfuric acid (DAP), 0.4~1.2% (w/v) NaOH (DALP) and 0.8~4.8% (w/v) FeCl_3_ solutions (FCP), respectively, based on a solid loading rate of 10% [7,22,23,27]. The above samples for all three pretreatments were then autoclaved at 121 °C for 30 min [7,22,23,27]. After the pretreated samples were centrifuged, the supernatants were collected and stored at –20 °C for further analysis. The solid residues were washed with deionized water until the filtrates were neutral. The solids were then dried in an oven at 105 °C to a constant weight. The dried solids were sealed in plastic bags and stored in a desiccator at room temperature until composition analysis and/or enzymatic hydrolysis.

### 3.3. Enzymatic Hydrolysis

The cellulase used for enzymatic hydrolysis were donated by Hunan Youtell Biochemical Co., Ltd. The protein content of the cellulase was 35 mg/mL. The activities of cellulase, β-glucosidase and endoglucanase were 30 FPU/mL, 6.8 U/mL and 165 U/mL, respectively. The enzymatic hydrolysis was performed in a 250-mL conical flask using 50-mM sodium acetate buffer (pH 5) containing 40 μL tetracycline hydrochloride with 2.5% solid loading at 50 °C and a 150-rpm agitation rate for 72 h. The enzyme was loaded at 15 FPU/g for the untreated and pretreated samples (i.e., 17.5 mg enzyme protein/g solids). One unit of cellulase activity is defined as the amount of the enzyme that releases 1 μmol of glucose per minute in the reaction mixture at 50 °C and pH 5. After solid–liquid separation via centrifugation, reducing sugars in the hydrolysate were analyzed by the 3, 5-dinitrosalicylic acid (DNS) assay [57]. All experiments were carried out in triplicate and all values were the means of triplicate ± SD.

The biomass conversion to fermentable sugar was calculated using the equation [19,47]:Conversion ratio (%) = 100 × Y_TRS_/(1.111 × C_C_ +1.136 × C_HC_)(1)
where Y_TRS_ (g/g pretreated sample) is the total reducing sugar yield per gram of pretreated sample in the enzymatic hydrolysate; the constants 1.111 and 1.136 are the conversion factors for cellulose/hemicellulose to the equivalent reducing sugars; and C_C_ and C_HC_ (g/g pretreated sample) are the cellulose and hemicellulose contents per gram of pretreated sample.

### 3.4. Scanning Electron Microscopy (SEM) Observation

The stalk samples were used to observe cell destruction before and after different pretreatments via scanning electron microscopy (SEM). The dried samples were fixed on a specimen holder with aluminium tape, then sputtered with gold in a JEOL JEC-1200 sputter-coater (Tokyo, Japan). All specimens were examined with a JEOL JSM-5600 LV scanning electron microscope (Tokyo, Japan) under high vacuum and at an accelerating voltage of 5.0 kV (10 µm, 500× magnification).

### 3.5. Analytical Methods

The sample mixtures were centrifuged and separated after pretreatments. The supernatants were collected to analyze total soluble sugar content. The solid residues were dried, and solid yields were recorded. Total solid and total soluble sugars were analyzed according to the standard methods [57,58,59]. The cellulose, hemicellulose and lignin contents were determined according to the standard method of Goering and Van-Soest [60], and calculated on the basis of residual total solids after pretreatments. All experiments were performed in triplicate, and all values were the means of triplicate ± SD.

Pretreatment severity factor (log *R_0_*) was calculated with the following equation [61,62]:*R_0_* = t*exp[(T_r_-100)/14.75](2)
where T_r_ is the reaction temperature (°C), 100 is the reference temperature (°C) and t is the reaction time (min). The fitted value (14.75) is the arbitrary constant *ѡ*.

## 4. Conclusions

Results in this study show that DAP, DALP and FCP resulted in the obvious structural changes and a high degree of hemicellulose and/or lignin removal from *P. alopecuroides* samples. Among all pretreatment methods tested, FCP produced the highest soluble sugar recovery (200.2 mg/g raw stalk) at the pretreatment stage. In comparison with FCP and DAP, DALP offered a much higher delignification and stronger morphological changes, which could significantly enhance the accessibility of the pretreated stalks to the enzymes, and thereby improve the performance of enzymatic hydrolysis. DALP gave the highest total soluble sugar yield of the pretreatment enzymatic-hydrolysis process (508.5 mg/g raw stalk). These results indicate that *P. alopecuroides*, a popular grass in huge quantities in China, could be used as a promising feedstock for biofuel production.

## Figures and Tables

**Figure 1 molecules-24-01715-f001:**
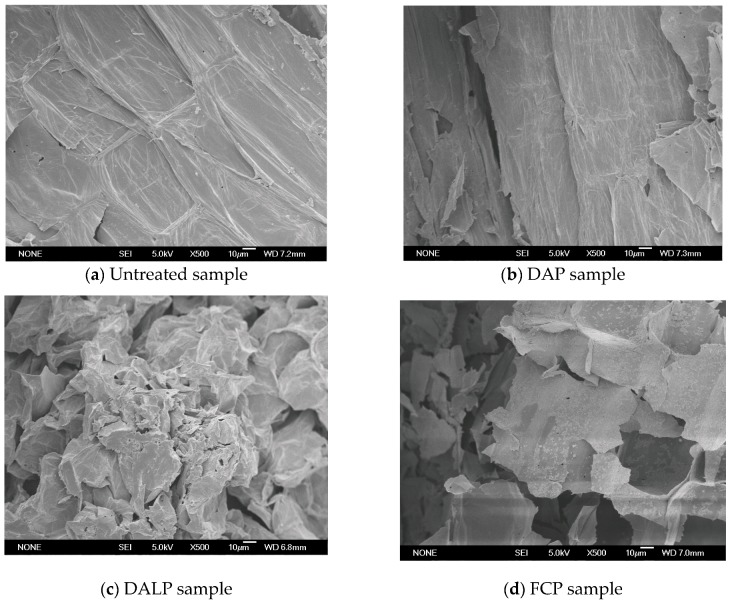
SEM images of *P. alopecuroides* samples with and without pretreatments (500×): (**a**) untreated sample; (**b**) sample with DAP; (**c**) sample with DALP; (**d**) sample with FCP.

**Figure 2 molecules-24-01715-f002:**
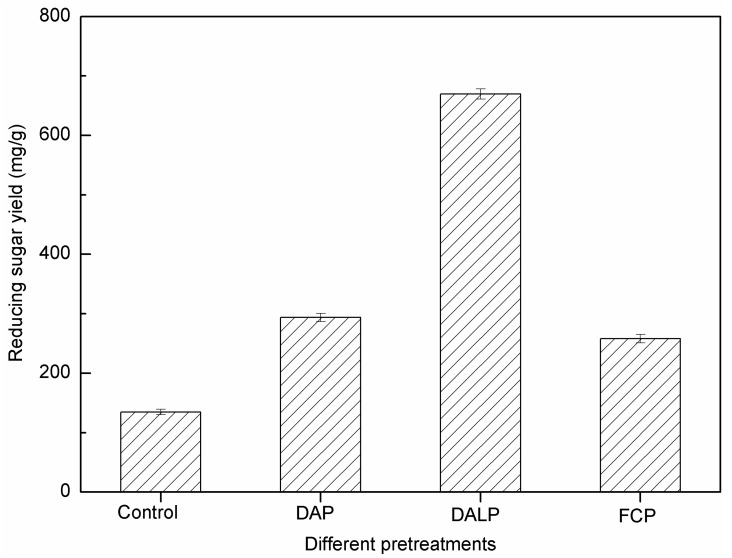
Effect of DAP, DALP and FCP on enzymatic hydrolysis.

**Figure 3 molecules-24-01715-f003:**
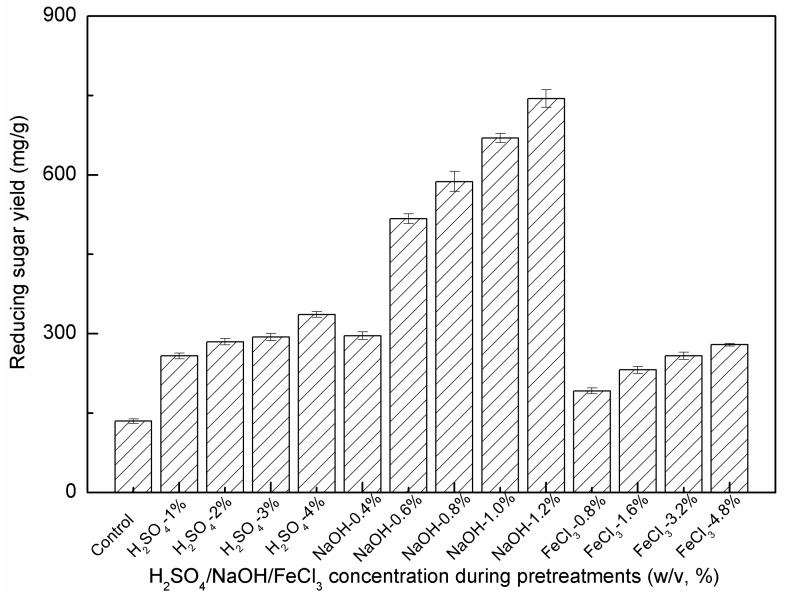
Effects of H_2_SO_4_/NaOH/FeCl_3_ concentrations during pretreatments on enzymatic hydrolysis.

**Table 1 molecules-24-01715-t001:** Effect of DAP/DALP/FCP of *Pennisetum alopecuroides* biomass on its chemical composition.

Different Pretreatment	Solid Yield (%)	Hemicellulose Content (%)	Cellulose Content (%)	Lignin Content (%)	Soluble Sugar from Pretreatment (mg/g RS) ^1^
Control	--	28.7 ± 0.4	41.8 ± 0.9	17.5 ± 0.6	20.0 ± 0.4
DAP, 3% H_2_SO_4_	53.3 ± 1.7	15.0 ± 1.7	55.9 ± 2.3	17.1 ± 1.5	112.2 ± 2.0
DALP, 1.0%NaOH	58.2 ± 1.2	15.3 ± 0.1	64.1 ± 1.7	11.7 ± 0.6	86.7 ± 0.2
FCP, 3.2% FeCl_3_	55.9 ± 1.8	11.9 ± 1.3	60.8 ± 1.3	16.4 ± 0.7	193.4 ± 8.7

^1^The soluble sugar yield in the pretreatment process was calculated based on per g raw stalk (RS). Values are means of triplicate ± standard deviation.

**Table 2 molecules-24-01715-t002:** The effects of H_2_SO_4_/NaOH/FeCl_3_ concentrations during pretreatments of *P. alopecuroides* on its chemical composition.

Different Pretreatment	H_2_SO_4_/NaOH/FeCl_3_ Concentrations (%)	Solid Yield (%)	Hemicellulose Content (%)	Cellulose Content (%)	Lignin Content (%)	Soluble Sugar from Pretreatment (mg/g RS) ^1^
Control	--	--	28.7 ± 0.4	41.8 ± 0.9	17.5 ± 0.6	20.0 ± 0.4
DAP with H_2_SO_4_	1%	59.4 ± 1.0	16.9 ± 1.6	51.8 ± 1.9	15.7 ± 1.1	40.0 ± 1.3
	2%	55.0 ± 0.9	15.5 ± 1.3	55.0 ± 0.5	16.8 ± 1.2	69.5 ± 1.2
	3%	53.3 ± 1.7	15.0 ± 1.7	55.9 ± 2.3	17.1 ± 1.5	112.2 ± 2.0
	4%	52.6 ± 2.1	12.2 ± 0.2	56.5 ± 1.6	17.3 ± 1.4	119.3 ± 0.7
DALP with NaOH	0.4%	74.6 ± 2.5	22.0 ± 0.9	53.3 ± 0.3	14.5 ± 0.4	50.5 ± 0.2
	0.6%	64.7 ± 1.7	17.4 ± 0.9	59.0 ± 1.4	14.0 ± 0.1	63.4 ± 2.0
	0.8%	59.4 ± 2.1	16.0 ± 0.2	63.2 ± 0.6	12.6 ± 0.1	83.5 ± 0.7
	1.0%	58.2 ± 1.2	15.3 ± 0.1	64.1 ± 1.7	11.7 ± 0.6	86.7 ± 0.2
	1.2%	53.9 ± 1.7	14.8 ± 0.3	68.0 ± 0.7	10.2 ± 0.8	107.3 ± 0.6
FCP with FeCl_3_	0.8%	68.8 ± 3.1	15.2 ± 1.7	56.9 ± 1.6	18.0 ± 1.6	90.2 ± 2.2
	1.6%	60.8 ± 1.7	13.5 ± 1.5	60.7 ± 1.1	17.3 ± 0.1	163.6 ± 3.2
	3.2%	55.9 ± 1.8	11.9 ± 1.3	60.8 ± 1.3	16.4 ± 0.7	193.4 ± 8.7
	4.8%	53.1 ± 2.3	9.7 ± 1.7	62.7 ± 1.5	15.3 ± 1.5	200.2 ± 6.7

^1^ The soluble sugar yield in the pretreatment process was calculated based on per g raw stalk (RS). Values are means of triplicate ± standard deviation. RS: raw stalk.

**Table 3 molecules-24-01715-t003:** Mass balance of the untreated and pretreated stalks.

Different Methods	Solid Yield (%)	Soluble Sugar from Pretreatment (mg/g RS)	Soluble Sugar from Enzymatic Hydrolysis (mg/g PS)	Soluble Sugar from Enzymatic Hydrolysis (mg/g RS)	Total Soluble Sugar Yield (mg/g RS) ^1^
Control	--	--	--	134.8	134.8
DAP	52.6	119.3	336.4	176.9	296.2
DALP	53.9	107.3	744.4	401.2	508.5
FCP	53.1	200.2	279.3	148.4	348.6

^1^ The total soluble sugar yield through the whole process from pretreatment to enzymatic hydrolysis was calculated based on per g raw stalk. RS: raw stalk; PS: pretreated stalk.

**Table 4 molecules-24-01715-t004:** Comparison of fermentable sugar recovery from different biomass.

Biomass	Pretreatment Conditions	Sugar Yield (mg/g PS) ^1^	Conversion Ratio (%)	Ref.
*P. alopecuroides*	1.2% NaOH, 121 °C, 30 min	744.4	85.4	This study
Wild rice grass	2% H_2_SO_4_, 121 °C, 60 min	457	93.2	[19]
Bamboo	1% NaOH, 3% Tween 80,121 °C, 60 min	629	75.4	[46]
Bamboo	1% H_2_SO_4_, 3% Tween 80,121 °C, 60 min	153	24.7	[46]
Pine foliage	1% C-TAB, 1% H_2_SO_4_, 121 °C, 60 min	588	98.1	[20]
Pine foliage	1% PEG-6000, 1% NaOH, 121 °C, 60 min	477	88.4	[20]
Eucalyptus	12.5% [TBA][OH], ultrasound irradiation (at a power of 360 W for 60 min)	426.6	51.5	[34]
Eucalyptus	2% NaOH, ultrasound irradiation (at a power of 360 W for 60 min)	362.3	56.6	[34]
*P. purpureum Schum*	0.5% NaOH, 90 °C, 60 min	(glucose yield: 245 mg/g RS)	NA	[9]
*P. purpureum*	1.5% NaOH, 121 °C, 60 min	146.9	24.7	[17]
*P. purpureum*	2% Ca(OH)_2_ or NaOH, 121 °C, 60 min	324~537^2^	65.5~88.7	[47]

^1^ Soluble sugar yields were calculated based on per g pretreated stalk. ^2^ Sugar yields and conversion ratios were calculated on the basis of the reported data of the cellulose/hemecellulose contents and glucose/xylose/reducing sugar yields in the corresponding references. RS: raw stalk; PS: pretreated stalk; Ref.: references; NA: not available.

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
