# Peer review of "Enhanced Enzymatic Hydrolysis of Pennisetum alopecuroides by Dilute Acid, Alkaline and Ferric Chloride Pretreatments"

_molecules, 2019, doi:10.3390/molecules24091715_

Round 1
Reviewer 1 Report
This manuscript addresses the effects of different pretreatment on Pennisetum alopecuroides and their enzymatic hydrolysis. This is somehow interesting for readers of Molecules journal; however, it is not enough to be accepted to the journal. The reviewer has the multiple queries listed below to improve the manuscript that should be replied in an exhaustive and sound basis before considering the article for publication.
1) The reviewer can see some minor errors in typos, grammatical errors, sentence structures, and font size (mainly figures) in the whole manuscript that should be corrected.
2) Introduction part is not enough to provide the current issue, limitation, approach, novelty of the current work, and key findings here. For example, the authors have described there are various pretreatment methods; however, there is no specific reasons and reasons why the authors have chosen this biomass, pretreatment condition, advantage, and disadvantage. The reviewer recommends that the authors need to provide more content and description in Introduction section.
3) Result section is nicely summarized and compared with three different pretreatment process and its enzymatic hydrolysis. However, the reviewer believes that many undesirable inhibitors such as acetic acid, HMF, furfural, phenolic molecules, and others during pretreatment process that should be described in the result section. Since phenols are powerful inhibitors for the enzyme, which can deactivate/inactivate enzyme activity, it should be described with possible inhibition after pretreatment. Please see below publications that will be helpful for review:
Kim, D., 2018. Physico-chemical conversion of lignocellulose: inhibitor effects and detoxification strategies: A Mini Review. Molecules, 23, 309.
Ximenes, E., Kim, Y., Mosier, N., Dien, B., Ladisch, M., 2011. Deactivation of cellulases by phenols. Enzyme. Microb. Technol. 48, 54-60.
Ximenes, E., Kim, Y., Mosier, N., Dien, B., Ladisch, M., 2010. Inhibition of cellulases by phenols. Enzyme. Microb. Technol. 46, 170-176.
4) The information of enzyme for hydrolysis is not enough; it is required the protein concentration (mg protein/ml), other enzyme activities such as FPU, endo-glucanase, and beta-glucosidase, lot number of enzyme, and other information. More information should be provided to someone who wants to follow the current work.
5) For enzyme loading, another unit (mg enzyme protein/g solids) is necessary to be compared with other references.
6) The reviewer wonders why the authors used DNS method for measuring the glucose concentration. It can measure the glucose concentration; however, it could not provide an exact number of concentration compared to HPLC. The reviewer believes that HPLC analysis with other sugars (cellobiose, arabinose, sucrose, xylose) and other potential inhibitors (acetate, HMF, furfural, and phenols) are required.
Author Response
Response to Reviewer 1 Comments
This manuscript addresses the effects of different pretreatment on Pennisetum alopecuroides and their enzymatic hydrolysis. This is somehow interesting for readers of Molecules journal; however, it is not enough to be accepted to the journal. The reviewer has the multiple queries listed below to improve the manuscript that should be replied in an exhaustive and sound basis before considering the article for publication.
Point 1: The reviewer can see some minor errors in typos, grammatical errors, sentence structures, and font size (mainly figures) in the whole manuscript that should be corrected.
Response 1: Thank you for your comments. The manuscript has been rechecked for typos, grammatical errors, sentence structures and font size (Figure 2 and 3).
Point 2: Introduction part is not enough to provide the current issue, limitation, approach, novelty of the current work, and key findings here. For example, the authors have described there are various pretreatment methods; however, there is no specific reasons and reasons why the authors have chosen this biomass, pretreatment condition, advantage, and disadvantage. The reviewer recommends that the authors need to provide more content and description in Introduction section.
Response 2: Thank you for your comment. More information has been added in Introduction section to summarize research progress and clarify the novelty of the current work. The Introduction has been revised as following:
1) more details about the biomass (e.g. annual yield data) was added to indicate our reasons for choosing it,
2) advantages, disadvantages and research progress of different pretreatments were summarized, and
3) the objective and novelty of the present study were also addressed (Line 43-96).
Point 3: Result section is nicely summarized and compared with three different pretreatment process and its enzymatic hydrolysis. However, the reviewer believes that many undesirable inhibitors such as acetic acid, HMF, furfural, phenolic molecules, and others during pretreatment process that should be described in the result section. Since phenols are powerful inhibitors for the enzyme, which can deactivate/inactivate enzyme activity, it should be described with possible inhibition after pretreatment. Please see below publications that will be helpful for review:
Kim, D., 2018. Physico-chemical conversion of lignocellulose: inhibitor effects and detoxification strategies: A Mini Review. Molecules, 23, 309.
Ximenes, E., Kim, Y., Mosier, N., Dien, B., Ladisch, M., 2011. Deactivation of cellulases by phenols. Enzyme. Microb. Technol. 48, 54-60.
Ximenes, E., Kim, Y., Mosier, N., Dien, B., Ladisch, M., 2010. Inhibition of cellulases by phenols. Enzyme. Microb. Technol. 46, 170-176.
Response 3:
Thank you for your comment. In this study, dilute acid, alkaline and ferric chloride pretreatments were investigated because these methods play important roles in removing lignin and hemicellulose, inducing structure changes on biomass surface, consequently enhancing digestibility of biomass. Previous studies reported that different pretreatment methods may produce undesirable inhibitors [1-5]; however, as demonstrated by our results, tested pretreatment strategies were still effective because cellulase was not inhibited in subsequent enzymatic hydrolysis of treated biomass. Data for reducing sugar yield (Figure 3) also confirmed the potential of these chemical pretreatment methods. This less severe inhibiting effect could be mainly attributed to the mild operation conditions used in this study (low acid/alkaline concentrations, 120oC), which would lead to lower concentration of by-product formed as compared to pretreatments operated at more severe conditions (160~260 oC) [1-5]. A brief discussion about the by-products has been added (Line 232-242). In future work, the effects of inhibitors on enzyme activity and microorganisms will be carefully addressed.
Kim, D. Physico-chemical conversion of lignocellulose: inhibitor effects and detoxification strategies: A Mini Review. Molecules, 2018, 23, 309.
Ximenes, E.; Kim, Y.; Mosier, N.; Dien, B.; Ladisch, M. Deactivation of cellulases by phenols. Enzyme. Microb. Technol. 2011, 48, 54-60.
Ximenes, E.; Kim, Y.; Mosier, N.; Dien, B.; Ladisch, M. Inhibition of cellulases by phenols. Enzyme. Microb. Technol. 2010, 46, 170-176.
Pandey, A.K.; Negi, S. Impact of surfactant assisted acid and alkali pretreatment on lignocellulosic structure of pine foliage and optimization of its saccharification parameters using response surface methodology. Bioresour. Technol. 2015, 192 , 115-125.
McIntosh, S.; Vancov, T. Optimisation of dilute alkaline pretreatment for enzymatic saccharification of wheat straw. Biomass Bioenergy 2011, 35, 3094-3103.
Point 4: The information of enzyme for hydrolysis is not enough; it is required the protein concentration (mg protein/ml), other enzyme activities such as FPU, endo-glucanase, and beta-glucosidase, lot number of enzyme, and other information. More information should be provided to someone who wants to follow the current work.
Response 4: Thank you for your comment. More information of enzyme for hydrolysis has been provided in Materials and Methods section, including filter paper activity, endo-glucanase and beta-glucosidase (Line 349~351).
Point 5: For enzyme loading, another unit (mg enzyme protein/g solids) is necessary to be compared with other references.
Response 5: Thank you for your comment. Another unit for enzyme loading has been added in Materials and Methods section (Line 355).
Point 6: The reviewer wonders why the authors used DNS method for measuring the glucose concentration. It can measure the glucose concentration; however, it could not provide an exact number of concentration compared to HPLC. The reviewer believes that HPLC analysis with other sugars (cellobiose, arabinose, sucrose, xylose) and other potential inhibitors (acetate, HMF, furfural, and phenols) are required.
Response 6: Thank you for your comment. In this work, our main purpose is to recover as many fermentable reducing sugars from pretreated biomass as possible, and the determination of total reducing sugar yield, which is commonly used as an indicator for enzymatic hydrolysis efficiency in previous studies [1-4], is hence sufficient. It is well-known that DNS method is a reliable and most widely accepted method for the determination of reducing sugar concentration [1-5]. Although exact number of concentration of different sugars can be measured by using HPLC, this method is complicated, expensive and time-consuming. Given the number of experimental conditions performed, DNS allowed us to conduct substantial experiments with limited funding. In future work, the concentration of glucose, xylose and other fermentable sugars will be carefully measured in sample treated under optimum DALP and/or other pretreatment condition, thereby providing more information for the application in subsequent ethanol fermentation. In addition, discussion about potential inhibitors and their possible effects have been added (Please refer to R3 to reviewer 1).
Wang, Z.N.; Hou, X.F.; Sun, J.; Li, M.; Chen, Z.Y.; Gao, Z.Z. Comparison of ultrasound-assisted ionic liquid and alkaline pretreatment of Eucalyptus for enhancing enzymatic saccharification. Bioresour. Technol. 2018, 254, 145-150.
Sahoo, D.; Ummalyma, S.B.; Okram, A.K.; Pandey, A.; Sankar, M.; Sukumaran, R.K. Effect of dilute acid pretreatment of wild rice grass (Zizania latifolia) from Loktak Lake for enzymatic hydrolysis. Bioresour. Technol. 2018, 253, 252-255.
Li, K.N.; Wan, J.M.; Wang, X.; Wang, J.F.; Zhang, J.H. Comparison of dilute acid and alkali pretreatments in production of fermentable sugars from bamboo: Effect of Tween 80. Ind. Crops Products 2016, 83, 414-422.
Pandey, A.K.; Negi, S. Impact of surfactant assisted acid and alkali pretreatment on lignocellulosic structure of pine foliage and optimization of its saccharification parameters using response surface methodology. Bioresour. Technol. 2015, 192 , 115-125.
Miller, G.L. Use of dinitrosalicylic acid reagent for determination of reducing sugar. Anal. Chem. 1959, 31, 426-428.

Reviewer 2 Report
This paper, entitled Enhanced Enzymatic Hydrolysis of Pennisetum alopecuroides by Dilute Acid, Alkaline and Ferric Chloride Pretreatments, is a scholarly work and can increase knowledge on this field. The paper is well written and well related to existing literature. The abstract and keywords are meaningful.
I have specific and general comments:
- please provide more details about the choice of the biomass selected for this study.
- What is the amount of this feedstock? What is the potential recovery of energy as biofuel correlated to the amount of biomass potentially used for this purpose? The authors should discuss about the biofuel production from sugars obtained by this way.
- What about the feasability to transfer this technical solution at pilot scale or industrial scale due to the use of enzymes? Is it possible to reuse enzymes? Which strategy could be applied in this way?
- What is the cost of use of enzymes for such application? Please provide cost analysis of such application.
- Please provide comparison with other pretreatments? The authors should provide a deeper discussion and comparison with other similar pretreatments
- About the discussion and conclusion, the authors should provide data or discussion about economical, environmental and technical considerations for such application.
- Please provide an energy balance between consumed energy and potential energy recovery with biofuel productiuon from this biomass pretreatment
From my point of view, this paper is not fully acceptable for publication and requires minor revision and some amendments, according to the comments listed previously. I recommend the following decision: ACCEPT WITH MINOR REVISION.
Author Response
Response to Reviewer 2 Comments
This paper, entitled Enhanced Enzymatic Hydrolysis of Pennisetum alopecuroides by Dilute Acid, Alkaline and Ferric Chloride Pretreatments, is a scholarly work and can increase knowledge on this field. The paper is well written and well related to existing literature. The abstract and keywords are meaningful.
I have specific and general comments:
Point 1: please provide more details about the choice of the biomass selected for this study.
Response 1: Thank you for your comment. More details about the choice of biomass selected for this study has been added to the Introduction section. Please refer to R2 to reviewer 1 for details.
Point 2: What is the amount of this feedstock? What is the potential recovery of energy as biofuel correlated to the amount of biomass potentially used for this purpose? The authors should discuss about the biofuel production from sugars obtained by this way.
Response 2: Thank you for your comment. More information about the amount of the biomass as well as its potential recovery of energy has been added. In brief, the dry biomass yield of this feedstock can reach as high as 40~50 t/ha. Due to the limited farmland resources in China, planting energy crops on available marginal land is increasingly regarded as one of the most promising choices for the production of biofuel feedstocks. If 20% of the marginal land area is used for planting Pennisetum alopecuroides, theoretically about 770 million tons of cellulosic ethanol can be produced annually. The corresponding revision can be seen from line 48~52 and 325~331 in the revised MS.
Point 3: What about the feasibility to transfer this technical solution at pilot scale or industrial scale due to the use of enzymes? Is it possible to reuse enzymes? Which strategy could be applied in this way?
Response 3: Thank you for your comment. The techno-economic feasibility of this process depends on the subsequent bio-energy products being selected. If we integrate this process into anaerobic digestion for biogas production, it is relatively feasible because DALP can be further optimized to reduce its energy cost and the pretreated mixtures can be directly converted into biogas by the selected mixed cultures in anaerobic digestion without extra addition of cellulase. If the current process is used for bioethanol fermentation, more extensive works are needed to further improve the whole process partially due to high cost of cellulase and low efficiency of hydrolysis-fermentation. Please refer to R6 to reviewer 2 for details of the discussion about techno-economic analysis of this process.
The enzymes can be reused by immobilization methods. Efforts on mechanism investigation and process optimizations in the whole process (e.g. more efficient cellulase, enzyme and cell immobilization for reusing, direct microbial conversion, strain selection and adaptation for improving efficiency, substrate/product inhibition and utilizing mixed substrates, integrated process development and detoxification of substrates) may result in more encouraging results. The corresponding revision can be seen from line 302~324 in the revised MS.
Point 4: What is the cost of use of enzymes for such application? Please provide cost analysis of such application.
Response 4: Thank you for your comment. In the case of lignocellulosic ethanol production, cellulase cost was about $0.5 per gallon ethanol, accounting for 20-30% of total costs. Given the increase of pretreatment performance and cellulase activity based on the continuous efforts of researchers from different universities, institutes and companies, we can increase enzymatic hydrolysis efficiency, thereby downstream process cost could be reduced. Please refer to R6 to reviewer 2 for details of the discussion about techno-economic analysis of this process. The corresponding revision can be seen from line 302~324 in the revised MS.
Point 5: Please provide comparison with other pretreatments? The authors should provide a deeper discussion and comparison with other similar pretreatments
Response 5: Thank you for your comment. Discussion about similar pretreatments has been enhanced as follows: 1) the enzymatic hydrolysis efficiencies of the pretreated samples were compared to indentify a potential pretreatment strategy for Pennisetum alopecuroides, and 2) data from different pretreatments were compared and discussed to illustrate the dominating factor in decreasing the biomass recalcitrance and thereby enhancing bioconversion. The corresponding revision can be seen from line 243~264 in the revised MS.
Point 6: About the discussion and conclusion, the authors should provide data or discussion about economical, environmental and technical considerations for such application.
Please provide an energy balance between consumed energy and potential energy recovery with biofuel productiuon from this biomass pretreatment
Response 6: Thank you for your comment. As responded in R3 to reviewer 2, the techno-economic feasibility of this process depends on the subsequent bio-energy products being selected. It will be much easier for future large scale application when this pretreatment process is integrated with biogas production. However, its application in bioethanol should also be carefully evaluated, considering the diversity of bioenergy. Notably, results in recent techno-economic analysis show that the cost of bio-ethanol from lignocellulosic biomass like Pennisetum alopecuroides are within the range of $0.50 to $0.65 per liter ethanol (about $1.91~$2.46 per gallon), which come close to the market price of ethanol ($2.50~$3.10 per gallon). A lower cost value is expected considering improved efficiency and the production of high-value chemicals. Current results indicate that lignocellulosic bioethanol from biomass like Pennisetum alopecuroides, as a green and sustainable alternative to gasoline, is gradually coming close to large industrial applications upon technical developments. A detailed discussion about the energy balance, techno-economic analysis, environmental benefit of this process can be seen from line 302~331 in the revised MS.

Round 2
Reviewer 1 Report
The revised version of manuscript has been improved and the authors clearly replied to the reviewers questions and suggestions. The reviewer has a couple of comments to improve the current manuscript that:
1) It would be much better to add a severity factor for the pre-treatment condition with temperature and time that would be a great thing to be discussed with the previous papers.
2) Table 4: a conversion ratio (%) of other biomass needs to be filled in, if the authors could not provide the ratio, it can be added with memo.
The reviewer really enjoyed to review this manuscript.
Author Response
Dear editor/Reviewers,
Thank you very much for your letter and for editing the MS (ID: molecules-491511). We would like to thank the reviewers for the time and effort they have spent reviewing our paper. The manuscript has been revised carefully based on their nice comments. All authors have seen the revised manuscript/response letter and approved to submit them to your journal.
In brief, pretreatment severity factor (log R0) was calculated. The corresponding method and results have been added in Materials and Methods as well as Discussion section, respectively. Data of conversion ratios in the cited references have been provided in Table 4. In addition, some errors in the MS have been corrected. All revisions have been done using Word’s “Track changes” function so that changes are easily visible.
Here below is our response to comments of all reviewers.
Thank you very much for your valuable time.
Sincerely,
Xiyu Cheng
Response to Reviewer 1 Comments
The revised version of manuscript has been improved and the authors clearly replied to the reviewers questions and suggestions. The reviewer has a couple of comments to improve the current manuscript that:
Point 1: It would be much better to add a severity factor for the pre-treatment condition with temperature and time that would be a great thing to be discussed with the previous papers.
Response 2: Thank you for your nice comment. Pretreatment severity factor (log R0) was calculated based on the previous method [1-2]. The method has been provided in Materials and Methods section (Line 383-387) and two references have been added (Line 576-579). The corresponding values of the pretreatment severity factor have also been used in Discussion section (Line 235-241).
[1] Overend, R.P.; Chornet, E.; Gascoigne, J.A. 1987. Fractionation of lignocellulosics by steam-aqueous pretreatments [and discussion]. Philos. Trans. R. Soc. Lond. A. 1987, 321, 523-536.
[2] Pedersen, M.; Meyer, A.S. Lignocellulose pretreatment severity-relating pH to biomatrix opening. New Biotechnol. 2010, 27, 739-750.
Point 2: Table 4: a conversion ratio (%) of other biomass needs to be filled in, if the authors could not provide the ratio, it can be added with memo.
Response 2: Thank you for your nice comment. Table 4 has been revised based on the comment (Line 300-304).
Other changes in MS
1. The amount of the marginal land area is not 550 million ha but 5.5 million ha [1]. This error has been corrected. The estimated potential yield of ethanol has also been corrected (Line 328-335).
[1] Chen, Y.Q.; Guo., X.D.; Liu, J.J.; Zhang, Z.J. Assessment of marginal land potential for energy plants in China. Land Develop. Eng. Res. 2017, 2, 1-7. http://dx.doi.org/CNKI:SUN:XBTG.0.2017-07-002 (In Chinese)
